# Custom-Made Titanium Mesh Tray for Mandibular Reconstruction Using an Electron Beam Melting System

**DOI:** 10.3390/ma14216556

**Published:** 2021-11-01

**Authors:** Isao Hoshi, Tadashi Kawai, Shingo Kurosu, Tadaharu Minamino, Kei Onodera, Ikuya Miyamoto, Hiroyuki Yamada

**Affiliations:** 1Division of Oral and Maxillofacial Surgery, School of Dentistry, Iwate Medical University, 19-1, Uchimaru, Morioka 020-8505, Iwate, Japan; isao.yamato.liz@gmail.com (I.H.); konodera@iwate-med.ac.jp (K.O.); ikuyam@iwate-med.ac.jp (I.M.); yamadah@iwate-med.ac.jp (H.Y.); 2Department of Elementary Material Process Technology, Iwate Industrial Research Institute, 2-4-25, Kitaiioka, Morioka 020-0857, Iwate, Japan; kurosu@pref.iwate.jp (S.K.); tada.minamino@gmail.com (T.M.)

**Keywords:** custom-made titanium mesh tray, electron beam additive manufacturing, mandibular reconstruction, optimization

## Abstract

Mandibular reconstruction using a titanium mesh tray and autologous bone is a common procedure in oral and maxillofacial surgery. However, there can be material problems—such as broken titanium mesh trays—which may undermine long-term functionality. This study was designed to investigate the optimal conditions for a titanium mesh tray with an ideal mandibular shape and sufficient strength, using computer-aided design, computer-aided manufacturing technology, and electron beam additive manufacturing. Specimens were prepared using Ti-6Al-4V extra low interstitial titanium alloy powder and an electron beam melting (EBM) system. The mechanical strength of the plate-shaped specimens was examined for differences in the stretch direction with respect to the stacking direction and the presence or absence of surface treatment. While evaluating the mechanical strength of the tray-shaped specimens, the topology was optimized and specimens with a honeycomb structure were also verified. Excellent mechanical strength was observed under the condition that the specimen was stretched vertically in the stacking direction and the surface was treated. The results of the tray-shaped specimens indicated that the thickness was 1.2 mm, the weight reduction rate was 20%, and the addition of a honeycomb structure could withstand an assumed bite force of 2000 N. This study suggests that the EBM system could be a useful technique for preparing custom-made titanium mesh trays of sufficient strength for mandibular reconstruction by arranging various manufacturing conditions.

## 1. Introduction

Reconstruction of the jaw is a necessary procedure for bone defects associated with tumor resection, trauma, or congenital diseases in oral and maxillofacial surgery [1,2]. In the mandible, if continuity is lost due to segmental resection, the masticatory function can be significantly impaired, and esthetic disorders due to mandibular deviation and facial deformity can occur [3]. For such mandibular defects, Dumbach reported mandibular reconstruction using a ready-made titanium mesh tray that mimics the morphology of the mandible and autologous iliac cancellous bone [4]. This is useful for linear defects in the area from the mandibular molars to the ramus of the mandible; however, compatibility between the mesh tray and the mandible can be poor in cases with complicated defects, or curved regions such as the chin [5]. To solve this problem, Tidstrom et al. reported a mandibular reconstruction method using a custom-made titanium mesh tray manufactured using a three-dimensional wax model, based on the information of several analyses [6]. In recent years, with the development of digital technology, it has become possible to create organs designed on a computer with a three-dimensional (3D) printer [7], a process which has also been applied to mandibular reconstruction [8,9].

Metal additive manufacturing methods can directly manufacture metal products using computer-aided design (CAD) data—many studies have been conducted on selective laser melting (SLM) technology [10,11], but electron beam melting (EBM) technology has also advanced [12,13]. At present, it is possible to manufacture products based on CAD data while freely designing complicated shapes, thicknesses, and strengths, which have been difficult to manufacture in the past [14,15,16]. Because CAD/computer-aided manufacturing (CAD/CAM) technology has the advantage that the thickness and strength of the metal can be designed [12,16], it should be able to reproduce the mandibular morphology of individual patients and manufacture customized titanium mesh trays of sufficient strength [17,18]. However, such additive manufacturing also has a problem in that an incomplete dissolution layer is formed on the outer surface, which affects mechanical strength [19]. Regarding mechanical strength, SLM products are reported to outperform or be equivalent to EBM products [20,21]. However, SLM products are responsible for lower crack thresholds, as compared to EBM products [21]. Therefore, there is a possibility that SLM products may break during manufacturing, and the probability of a breakage is expected to be enhanced if the form of the product is complex.

Although some custom-made titanium mesh trays for mandibular reconstruction have been reported, there are no reports on mechanical strength evaluation for three-dimensional products assuming occlusal force. In this study, to prepare a custom-made titanium mesh tray of sufficient strength using the EBM system and apply it to mandible reconstruction in the future, the materials of samples prepared using the EBM system were evaluated, as well as their optimal conditions.

## 2. Materials and Methods

### 2.1. Specimens

Specimens were prepared using an EBM system (Arcam EBM AX2, General Electric Company, Boston, MA, USA). Two types of samples were prepared: one stretching parallel to the stacking direction and the other stretching vertically (Figure 1a). Moreover, specimens with an untreated surface texture and those receiving an alumina sandblast treatment were prepared. From these, four types of specimens were prepared: those that were parallel to the stacking direction, with an untreated surface (PU); those that were parallel, with a treated surface (PT); those that were vertical to the stacking direction, with an untreated surface (VU); and those that were vertical, with a treated surface (VT). Extra low interstitial (ELI) Ti-6Al-4V (UAC065-170510 & ASTM F3001, Arcam EBM, Mölndal, Sweden) titanium alloy powder, with a particle size of 45–100 mm and an average particle size of 70 mm, was used for this study. The standard recipe recommended by the equipment manufacturer was used for the molding conditions of the specimens. A clinically used ELI Ti-6Al-4V commercial product (CP) (extended material, ASTM F136-13, VSMPO-AVISMA Corporation, Verkhnaya Salda, Russia) was used as a control.

### 2.2. Surface Characterization

Plate-shaped VU and VT specimens of length 70 mm, width 40 mm, and thickness 1.5 mm were prepared, and the surfaces of the specimens were observed using a digital microscope (VHX-7000, KEYENCE, Osaka, Japan). The CP was used as a control. The cross section of each specimen was observed after polishing using an automatic grinding machine (Buehler Beta grinder-polisher vector power head, Buehler, Lake Bluff, IL, USA). The surface roughness was measured using a DSF600 (Kosaka Laboratory Ltd., Tokyo, Japan), and the average roughness (Ra) and average peak-to-valley height (Rz) were calculated. Elemental analysis was performed using an electron microscope (JSM-7100F, JEOL, Tokyo, Japan).

### 2.3. Static Bending Test

Plate-shaped specimens of length 70 mm and width 40 mm were readied, with five types of thicknesses (1.0, 1.5, 2.0, 2.5, and 3.0 mm) being prepared for each of the PU, VU, PT, VT, and CP specimens. An Instron 8874 mechanical testing system (Instron, Norwood, MA, USA) was used for each bending test. Subsequently, a three-point bending test was conducted on each specimen, in accordance with the Japanese JIS T 0312 standard. Figure 1b shows a schematic of the three-point bending test. The distance between the loading roller and supporting rollers was set to 25 mm, and the diameters of the loading roller and supporting rollers were 10 mm and 4 mm, respectively. From the load–displacement curve obtained, the 0.2% offset load was evaluated under the condition of 0.05 mm for the offset displacement corresponding to the 0.2% displacement. The bending strength (M) could be calculated as M = *h* × *p*/2 (N·m), where *p* is the offset load (N), and *h* is the distance between the supporting rollers and the loading roller. The bending stiffness (EI) could be calculated as EI = A × *h*^3^/6 (N·m^2^), where A is the slope of the load–displacement curve.

### 2.4. Bending Fatigue Test

Plate-shaped specimens of length 70 mm, width 40 mm, and thickness 1.5 mm were prepared for each of the PU, PT, VU, VT, and CP specimens. The test conditions were as follows: the cyclic wave shape was sinusoidal, the load ratio was 0.1, the cyclic frequency was 3 Hz, and the number of cycles was the cumulative number of tests until the specimens were destroyed. When the number of cycles was 10^6^ or more, no breakage was defined. From the results, an L–N curve was generated; the vertical axis representing the maximum load (L) and the horizontal axis representing the number of cycles (N).

### 2.5. Bending Test of Tray-Shaped Specimens

Tray-shaped specimens of length 120 mm, height 32.0 mm, and width 12.1 mm were prepared. The stacking direction was set vertically to the lower edge of the specimen, and surface treatment was performed on the specimen (VT). The tray-shaped specimen was placed on a specific device of height 44.98 mm and width 12.1 mm, as shown in Figure 1c. Four specimen thicknesses were prepared—that is, 0.3, 0.6, 0.9, and 1.2 mm. Both ends of the specimen were fixed and a load (N) was applied to the center of the specimen placed on the base. The load was then increased to 5000 N. A load–displacement curve was generated and each specimen was evaluated using a 0.2% offset load value for the presence or absence of plastic deformation at 2000 N, which was assumed to be the bite force [22].

### 2.6. Topology Optimization of Tray-Shaped Specimens and Evaluation of Mechanical Strength

Tray-shaped VT specimens of thickness 1.2 mm were analyzed using commercial software (Altair OptiStruct™, Inspire, Altair Engineering, Ltd., Ann Arbor, MI, USA) and topologically optimized. The specimens were analyzed for 20%, 30%, 40%, and 50% volume fraction constraints, and the form of each specimen was designed. As a condition for topology optimization, the load was set to 2000 N, and the area excluding the fixed part was analyzed. Moreover, to maintain the function of the tray in the specimens obtained using topology optimization, a honeycomb structure was added to the excluded range and evaluated in the same way. Figure 1d shows an outline of the tray design.

### 2.7. Statistical Analysis

All values are reported as mean ± standard deviation. All experiments were performed at least six times, showing reliable reproducibility. All statistical analyses were performed for each experiment. One-way analysis of variance (ANOVA) was used to compare the means among the four groups using the EBM system. If the ANOVA was significant, Tukey’s multiple comparison analysis was used as a post hoc test. A *t*-test was performed to compare each specimen using the EBM system with the control.

## 3. Results

### 3.1. Surface Characterization

A digital microscope revealed that grooves were formed in a vertical direction to the stacking direction and that the metal particles on the surface were removed by the alumina sandblast treatment (Figure 2a,b). In the cross section, the rough surface of the specimen stretched parallel to the stacking direction was less than that of the specimen perpendicular to the stacking direction, with the alumina blasting treatment reducing the surface roughness (Figure 2d–g). Although there was no significant difference in the Ra results of the PU, PT, and VU specimens, the results of the VT specimen were significantly lower (Figure 3a). The Rz results indicated no significant difference between the PT and VU specimens. The Rz of the PU specimens was larger than that of the others, and the Rz of the VT specimen was smaller than that of the others (Figure 3b). Elemental analysis showed that the specimen surface with alumina sandblast treatment was rough in comparison to specimens with no treatment, also exhibiting the attachment of oxygen and aluminum elements to the alumina sandblast-treated surface (Figure 4).

### 3.2. Static Bending Test

At 0.2% offset load, the PU specimen values were significantly lower than those of the other samples at thicknesses of 1.0, 1.5, and 2.0 mm. The VT specimen value was significantly higher than that of specimens with a thickness of 1.5 mm, and the VU specimen value was higher than that of specimens with a thickness of 2.0 mm. The PT specimen values were significantly higher than those of the PU specimens under the condition of thicknesses of 2.5 and 3.0 mm. The CP value was higher than that of other specimens under the condition of thickness of 1.0, 1.5, and 2.0 mm (Figure 5a).

In the bending strength test, the PU specimen values were significantly lower than those of the other specimens at thicknesses of 1.0, 1.5, and 2.0 mm. The PU specimen value was significantly lower than that of the VU specimen with a thickness of 2.5 mm. The CP value was higher than that of the other specimens under the condition of thicknesses of 1.0, 1.5, and 2.0 mm (Figure 5b). 

For the bending stiffness, the PU specimen value was significantly lower than that of the other specimens, and the PT specimen value was significantly lower than that of the VT specimen under the condition of thickness of 1.0 mm. Under the condition of 1.5 mm thickness, the PU specimen value was significantly lower than that of the VU and VT specimens, and the PT specimen value was significantly lower than that of the VT specimen. The CP values were higher than those of the other specimens under all thickness conditions (Figure 5c).

### 3.3. Bending Fatigue Test

The number of integrations until breaking, under the same load, increased in the order of the PU, PV, VU, VT, and CP specimens. It was confirmed that the VT specimen did not break even when the number of integrations was 10^6^ or more at 300 N (fatigue limit). The fatigue limit of the CP was 900 N (Figure 6).

### 3.4. Bending Test of Tray-Shaped Specimens

Six samples were prepared for each thickness during the preparation of the tray-shaped specimens. The specimens designed with a thickness of 0.3 and 0.6 mm had different thicknesses in the actual measurement, which were 0.770 ± 0.009 and 0.802 ± 0.004 mm, respectively. The specimens designed and prepared to a thickness of 0.9 mm were almost 0.1 mm thicker than the design, which was 1.003 ± 0.010 mm. Conversely, the 1.2 mm thick specimens were practically the same as the design, which was 1.253 ± 0.004 mm. The average 0.2% offset load values designed to have a thickness of 0.3, 0.6, 0.9, and 1.2 mm were 385 ± 5.0, 1081 ± 25.6, 2162 ± 21.1, and 4163 ± 45.3 N, respectively. The plastic deformation with the load of 2000 N was more than 2.0 mm for the specimens designed to have a thickness of 0.3 and 0.6 mm. Little plastic deformation was observed in the 0.9 and 1.2 mm thick specimens at 2000 N (Figure 7).

### 3.5. Topology Optimization of Tray-Shaped Specimens and Evaluation of Mechanical Strength

In the 20% volume fraction constraint, the specimen quantity increased to approximately 50% (Figure 8a). The 0.2% offset load values of the specimens with a 30%, 40%, and 50% volume fraction constraint without the honeycomb structure were higher than 2000 N; however, that of the specimen with a 20% volume fraction constraint was 1806.7 ± 96.2 N. Conversely, the 0.2% offset load values of each specimen with the honeycomb structure increased significantly, exceeding 3000 N (Figure 8a). The average 0.2% offset load value of the specimen with a 20% volume fraction constraint was 3165.0 ± 125.3 N.

The bending strength increased significantly between 30% and 40% of the volume fraction constraint without the honeycomb structure (Figure 8b). The bending stiffness gradually increased with the volume fraction constraint (Figure 8c). The honeycomb structure increased both the bending strength and bending stiffness.

## 4. Discussion

Bone reconstructive surgery for bone defects in the mandible is one of the main treatments in the field of oral surgery. Reconstruction with a titanium mesh tray and autologous iliac cancellous bone, a fibula and a titanium plate, and a calcium phosphate material three-dimensionally prepared based on the morphology of the mandibular defect, has been reported in the past [23,24,25,26,27]. In this study, reconstruction with titanium mesh and autologous bone was assumed, and the purpose was to improve the certainty of mandible reconstruction by improving the material of the titanium mesh. Another group has reported the usefulness of titanium mesh, using the selective laser melting method for reconstruction of alveolar bone defects [17]; however, there has been no clinical application for large bone defects. In this study, it was expected that the titanium mesh manufactured using the EBM technique could be used for large bone defects by improving the production conditions.

Grooves were observed on the surface in the vertical direction to the stacking direction using a digital microscope, and this direction of the groove was expected to be related to strength. The alumina blasting treatment removed the metal particles, reducing Ra and Rz—that is, it was expected that the strength would increase as the stress concentration was reduced by smoothing the surface [21]. It was also thought that there could be a peeling effect due to the alumina blast treatment of the surface [19,28]. However, it was confirmed that oxygen and aluminum adhered to the sandblast-treated surface.

Since it is thought that this residue could cause damage to cells and surrounding tissues, surface treatment with other materials may be considered, or safety may be confirmed by in vivo examination in the future.

In the static bending test, the 0.2% offset load was larger in the vertical specimen than in the parallel specimen in the stacking direction. Moreover, the surface-treated specimen exhibited a larger result with a 2% offset load than the untreated specimen. 

In the bending strength test, the PU specimen result was significantly lower than the results of the other specimens of thicknesses 1.0, 1.5, and 2.0 mm. There was a significant difference between the PU and VU specimens of thickness 2.5 mm. 

In the bending stiffness test, the specimens vertical to the stacking direction were higher than those of the parallel specimens of thicknesses 1.0 and 1.5 mm. From these results, it is suggested that vertical to the stacking direction is the most suitable condition for a custom-made titanium mesh tray using EBM. No conditions were significantly better than those of the CP.

In the bending fatigue test, the cumulative number of cycles until the specimens were destroyed significantly increased using the alumina blast treatment. It was shown that the presence or absence of surface treatment had a significant influence on the bending fatigue characteristics. A comparison of the EBM specimens with the CP exhibited a difference of approximately three times. It is thought that an incomplete dissolution layer formed on the specimen surface by EBM has an influence. Consequently, to improve the bending fatigue characteristics, surface treatment appears to be a necessary condition.

The bite force quotient was mainly applied vertically to the mandible. The results of the mechanical strength test suggest that it is necessary to design the inferior border of the mandible as vertical to the stacking direction and to perform surface treatment in manufacturing custom-made titanium mesh trays.

The results of the bending test of tray-shaped specimens indicated that a thickness of at least 0.9 mm was needed to manufacture tray-shaped specimens because of the preparation process and strength. However, as the real thickness of the specimen designed to be 0.9 mm thick was almost 1.0 mm, a thickness of 1.2 mm was selected for the design of the tray-shaped specimen. As the tray-shaped specimen could not maintain its tray shape from the result of topology optimization of the 10% volume fraction constraint, the volume fraction constraint was set to more than 20%. The results of the 0.2% offset load value of the tray-shaped specimens prepared based on topology optimization demonstrated that the value of the specimen with the 20% volume fraction constraint was lower than 2000 N, which was assumed to be the bite force. However, the honeycomb structure could increase the value to more than 3000 N. Although the bending fatigue characteristic of the plate-shaped VT specimen was 300 N, which was approximately one-third of that of the CP, it was suggested that a tray-shaped VT specimen with a honeycomb structure could withstand the bite force.

Clinically, the honeycomb structure is effective for bone formation because it makes it easier to supply blood and cells concerned with bone formation [29]. However, because the 0.2% offset load value of just the honeycomb structure was almost 700 N (data not shown), some tray morphology is considered necessary for strength.

In this study, the optimal conditions for the preparation of a custom-made titanium mesh tray for mandibular reconstruction using an EBM system were examined. In order to resist bite force, it would be desirable for a custom tray to have a stacking direction vertical to the inferior border of the mandible and to have surface treatment. Furthermore, it was shown that the thickness was 1.2 mm, the weight reduction rate was 20%, and the addition of the honeycomb structure could resist the bite force. For clinical application, further improvement of the production conditions and experimental evaluation would be required in the future.

## 5. Conclusions

The results and findings of this study suggest that the custom-made titanium mesh tray using the EBM system can be sufficiently strong for mandibular reconstruction assuming occlusal force, given that various manufacturing conditions are specified. Such conditions include a stacking direction vertical to the lower edge of the mandible, a 1.2 mm thickness, a 20% volume fraction constraint, adding a honeycomb structure, and surface treatment.

## Figures and Tables

**Figure 1 materials-14-06556-f001:**
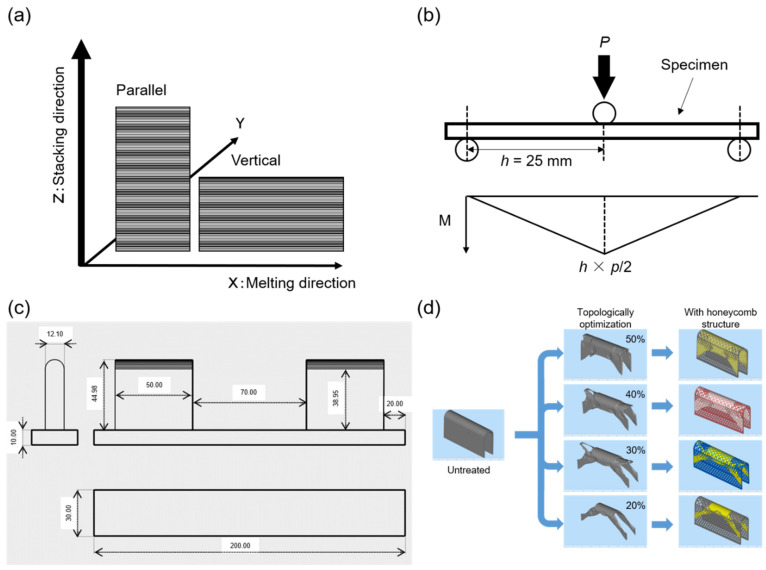
(**a**) Explanation of “parallel” and “vertical” to the stacking direction in specimen manufacturing. (**b**) Explanation of the three-point bending test. *p*: 0.2% offset load (N). *h*: distance between the loading roller and the supporting rollers (m). M: *h* × *p*/2 is the bending strength (N·m). (**c**) Design of a device for the bending test of tray-shaped specimens. *p*: load (N). Units in mm. (**d**) Design of tray-shaped specimens with topological optimization of 20%, 30%, 40%, and 50% volume fraction constraint. The right-hand side shows a design with a honeycomb structure added.

**Figure 2 materials-14-06556-f002:**
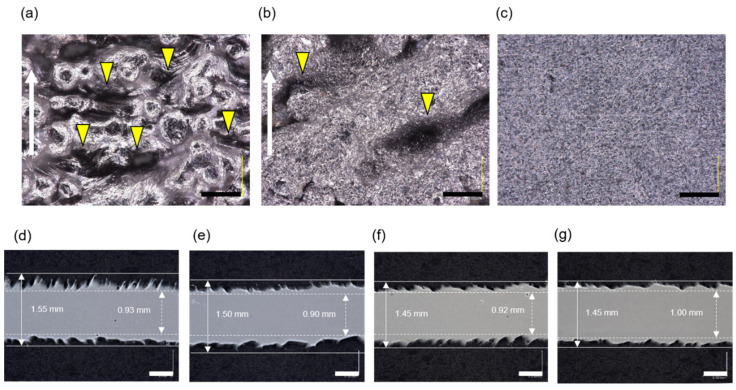
Digital microscope images of each specimen surface of (**a**) untreated, (**b**) alumina sandblast treatment, and (**c**) control. The white arrow indicates the stacking direction. The yellow triangles indicate the grooves. Bar = 100 mm. The grooves are formed in the vertical direction to the stacking direction, with the metal particles on the surface being removed using the alumina blasting treatment. Also shown are pictures of the cross section of the (**d**) PU, (**e**) VU, (**f**) PT, and (**g**) VT specimens. Bars = 0.5 mm. The surface of the specimen stretched parallel to the stacking direction is less rough than that of the specimen vertical to the stacking direction, as the alumina blasting treatment reduces the surface roughness.

**Figure 3 materials-14-06556-f003:**
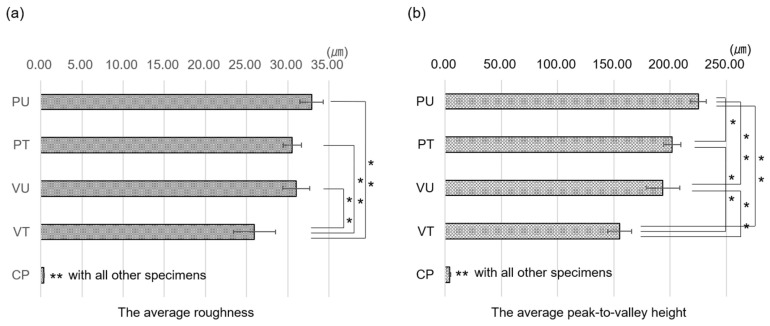
(**a**) Ra results. Although, there is no significant difference in the Ra results of the PU, PT, and VU specimens, the Ra result of the VT specimen is significantly lower. (**b**) Rz results. There is no significant difference between the PT and VU specimens. The Rz of the PU specimen is larger than that of the others, and that of the VT specimen is smaller than that of the others. * *p* < 0.05, ** *p* < 0.01.

**Figure 4 materials-14-06556-f004:**
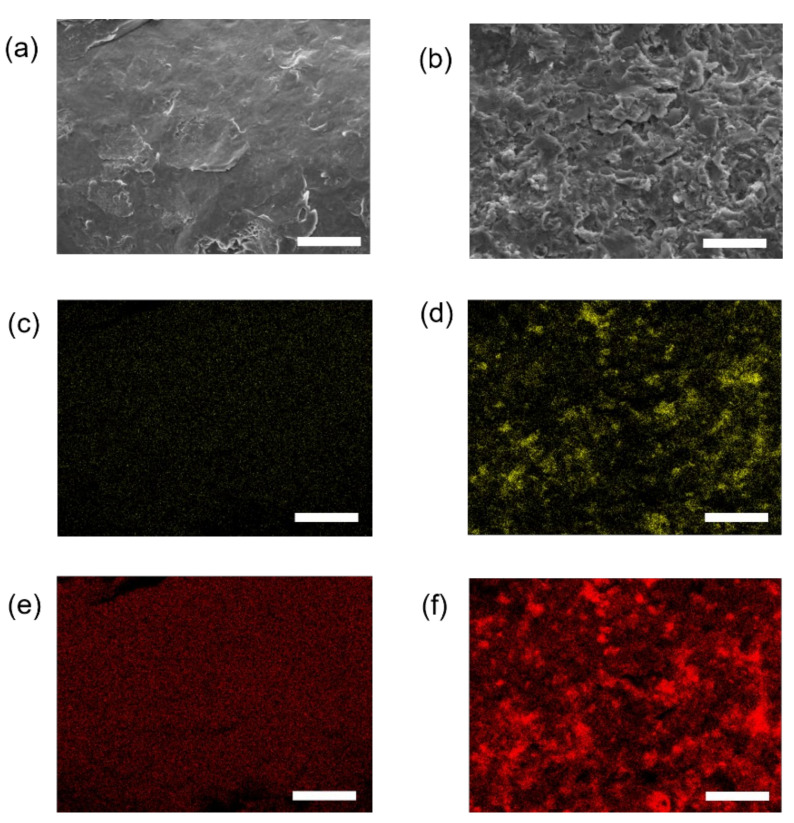
Images of specimen surfaces using electron microscopy with (**a**) no treatment and (**b**) alumina sandblast treatment. The specimen surfaces with alumina sandblast treatment were rough in comparison to those with no treatment. Images of oxygen elements on the surface of specimens (**c**) with no treatment and (**d**) with sandblast treatment. The yellow area indicates the presence of oxygen elements. Images of aluminum elements on the surface of specimens (**e**) with no treatment and (**f**) with sandblast treatment. The red area indicates the presence of aluminum elements. The attachment of oxygen and aluminum elements on the surface with alumina sandblast treatment are clearly shown. Bars = 25 μm.

**Figure 5 materials-14-06556-f005:**
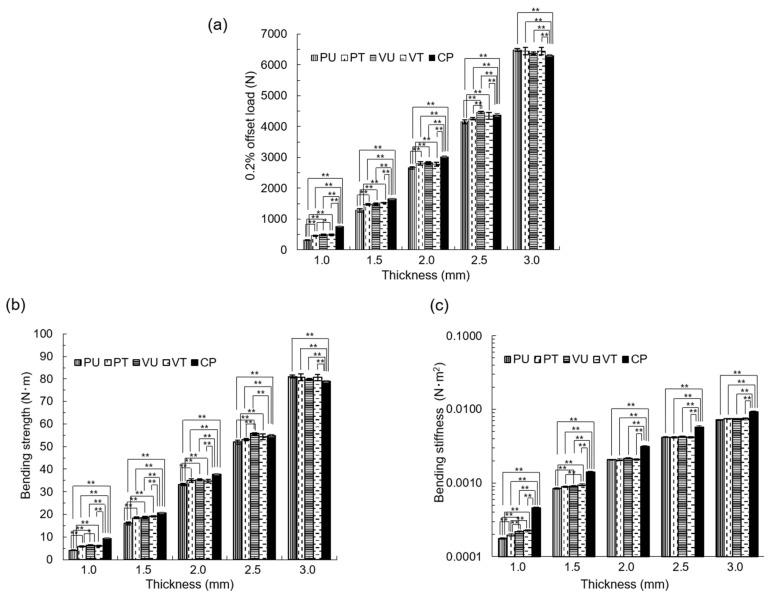
Static bending test results. (**a**) 0.2% offset load. (**b**) Bending strength. (**c**) Bending stiffness. * *p* < 0.05, ** *p* < 0.01.

**Figure 6 materials-14-06556-f006:**
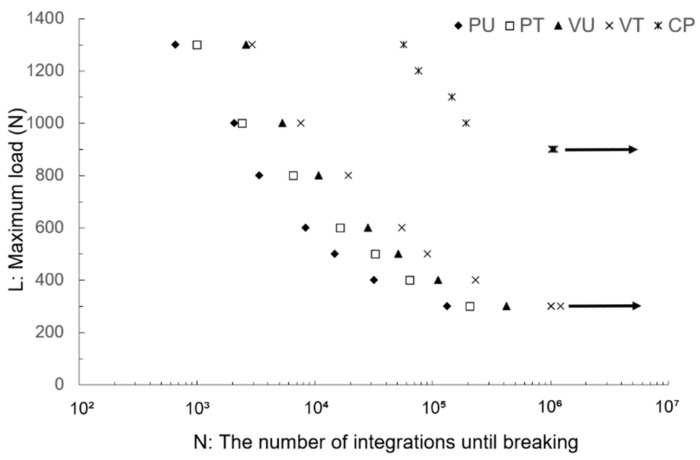
Bending fatigue test results. The L–N curve is shown. The VT specimens do not break even if the number of integrations is 10^6^ or more at 300 N (the fatigue limit). The arrows indicate no breakage.

**Figure 7 materials-14-06556-f007:**
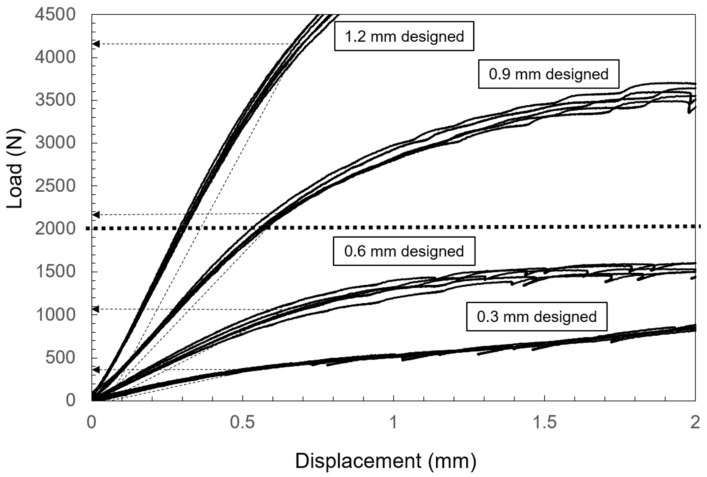
Load–displacement curve from the bending test of tray-shaped VT specimens. Little plastic deformation is observed in the 0.9- and 1.2 mm thick specimens at 2000 N. The dot arrows indicate the average of the 0.2% offset load. The dot line indicates 2000 N to be the assumed occlusal force.

**Figure 8 materials-14-06556-f008:**
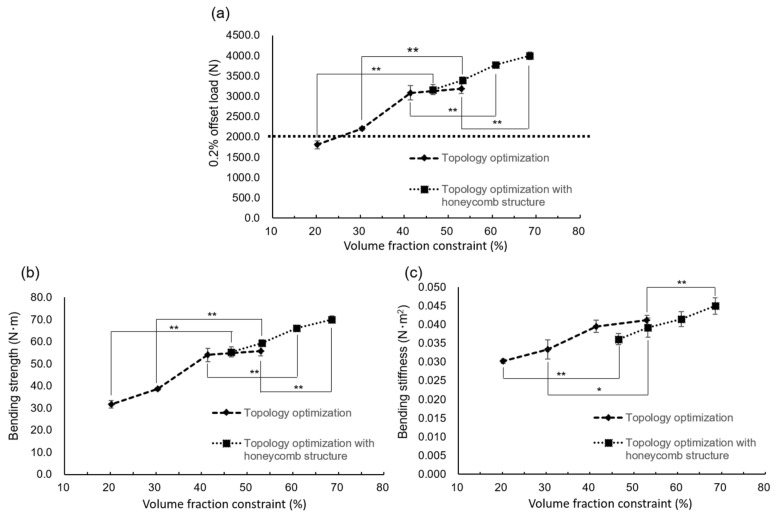
Mechanical strength test results for the 1.2 mm thick tray-shaped VT specimen. (**a**) 0.2% offset load. The dot line indicates 2000 N to be the assumed occlusal force. The 0.2% offset load values of each specimen with the honeycomb structure increase significantly to more than 3000 N. (**b**) Bending strength. (**c**) Bending stiffness. The honeycomb structure increases both the bending strength and bending stiffness. * *p* < 0.05, ** *p* < 0.01.

## Data Availability

The raw data required to reproduce these results cannot be shared at this time as the data also forms part of an ongoing study.

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
