# Peer review of "Custom-Made Titanium Mesh Tray for Mandibular Reconstruction Using an Electron Beam Melting System"

_materials, 2021, doi:10.3390/ma14216556_

Round 1

Reviewer 1 Report

The article subjected to custom-made titanium mesh tray for mandibular reconstruction using electron beam melting system. The paper could be interested for researchers who work with EBM for biomedical application. However some corrections should be done before the publication:
1)    Abstract should be written as single text without separation to numbered parts.
2)    Novelty should be described clearer.
3)    There are a lot of real application examples of SLM for personalized implants manufacturing. The authors should describe why they prefer EBM instead of SLM.
4)    Conclusions are not representative. Authors are welcome to add more specifics.

Author Response

Reviewer 1

  1. Thank you for your comment. I corrected as you pointed out.
  2. Thank you for your advice. The novelty of this study is the evaluation of the mechanical strength of the three-dimensional product using the EBM system assuming occlusal force. I added the sentence “Although some custom-made titanium mesh trays for mandibular reconstruction have been reported, there are no reports of evaluating the mechanical strength of the three-dimensional product assuming occlusal force.” before the sentence of “In this study, to prepare a custom-made titanium mesh tray of sufficient strength using the EBM system and apply it to mandible reconstruction in the future, the materials of samples prepared using the EBM system were evaluated, as well as their optimal conditions.”
  3. Thank you for your advice. There is a possibility that the SLM products may break during manufacturing due to its lower crack thresholds. I added some citations and the sentence “Regarding mechanical strength, the SLM products are reported to be higher or equivalent to EBM products [20, 21]. However, the SLM products are responsible for the lower crack thresholds, as compared to the EBM products [21]. Therefore, there is a possibility that the SLM products may break during manufacturing, and it is expected that the products will break even more if the form of the product is complicated.” in the introduction section.
  4. Thank you for your advice. I corrected the conclusion to “This study suggests that the custom-made titanium mesh tray using the EBM system can get sufficient strength for mandibular reconstruction assuming occlusal force by arranging various manufacturing conditions; such as the stacking direction is vertical to the lower edge of mandible, the thickness is 1.2 mm, the volume fraction constraint is 20%, the addition of honeycomb structure, and surface treatment.”

Reviewer 2 Report

- The approach is interesting and the topic is appropriate for the journal.

  • The work has a very clear structure and all the sections are well written in a way that is easy to read and understand.
  • Little modifications and improvements are needed to enhance the quality of the paper.

  • The paper deals with the Custom-Made Titanium Mesh Tray for Mandibular Reconstruction Using Electron Beam Melting System, reporting interesting results. In the “Introduction” section, the authors start to discuss about the reconstruction of the jaw and additive manufacturing techniques such as selective  laser melting technology and electron beam melting (EBM) technology. For this reason, even though the authors already report some works in the literature and they basically focus on EBM technology for ELI Ti-6Al-4V titanium alloy powder, I also suggest to BRIEFLY report the progresses in the design of  3D additive manufactured structures with tailored functionalities and properties for several applications (e.g., Journal of Healthcare Engineering, 2019, 2019, 3212594…). In general, the idea should be to further stress the potential of different additive manufacturing techniques in the design of devices with tailored and improved properties. Then,  the authors should focus on the aim of their work and on the specific approach,  reporting their study related Custom-Made Titanium Mesh Tray for Mandibular Reconstruction Using Electron Beam Melting System. All of this should improve the quality of the paper, reporting important features as well as further methodologies  and analysis in designing 3D additive manufactured devices tailoring the functional properties according to the specific applications, thus helping the different kinds of readers to better understand the value of their work.
  • The Introduction and/or discussion section as well as the list of references should be improved according to the above reported comments.
  • In all of the reported graphs in the Figures, on the y and x axes I suggest to report the units of measurement between two brackets (e.g., Load(), Displacement (), Strength ()…)
  • The quality of some figures should be improved.
  • The title is adequate and appropriate for the content of the article.
  • The abstract contains information of the article.
  • Figures and captions are essential and clearly reported.

Author Response

Reviewer 2

  1. Thank you for your advice. I added some citations and the sentences “Regarding mechanical strength, the SLM products are reported to be higher or equivalent to EBM products [20, 21]. However, the SLM products are responsible for the lower crack thresholds, as compared to the EBM products [21]. Therefore, there is a possibility that the SLM products may break during manufacturing, and it is expected that the products will break even more if the form of the product is complicated.” and “Although some custom-made titanium mesh trays for mandibular reconstruction have been reported, there are no reports of evaluating the mechanical strength of the three-dimensional product assuming occlusal force.” in the introduction section to improve the quality of this paper.
  2. I added the units of measurement in the y and x axes title and all figures are improved.

Reviewer 3 Report

Dear authors,

Your papers is brilliant and bring new insights into the oral and maxillofacial area. To be honest, I think this papers should be accepted. I have no further improvements to bring besides the low quality of the pictures.

Congratulations and I hope I will your innovation in my clinical practice.

Best regards

Figure 1-4: Bring them together and make 1 panel.

All figures: Please introduce another one, bigger and with a higher resolution (could be 600 DPI).

Author Response

Reviewer 3

  1. Thank you for your advice. Figures 1-4 are all combined into one figure, and all figures are improved.

Round 2

Reviewer 1 Report

The Manuscript can be published.